# 2.5D U-Net for abdominal multi-organ segmentation

Ruixiang Lei[1][0009−0008−7413−5860] , Mingjing Yang[1][✉]

Intelligent Image processing and Analysis Laboratory,Fuzhou University, Fuzhou FJ 350108,CHN yangmj5@fzu.edu.cn

**Abstract.** Accurate and efficient segmentation of multiple abdominal organs from medical images is crucial for clinical applications such as disease diagnosis and treatment planning. In this paper, we propose a novel approach for abdominal organ segmentation using the U-Net architecture. Our method addresses the challenges posed by anatomical variations and the proximity of organs in the abdominal region. To improve the segmentation accuracy, we introduce an attention mechanism into the U-Net architecture. This mechanism allows the network to focus on salient regions and suppress irrelevant background regions, enhancing the overall segmentation performance. Additionally, we incorporate 3D information by connecting three consecutive slices as 3-dimensional inputs. This enables us to exploit the spatial context across the slices while minimizing the increase in GPU memory usage. We evaluate our proposed method on the MICCAI FLARE 2023 validation dataset, the mean DSC is 0.3683 and the mean NSD is 0.3668.

**Keywords:** organ segmentation · U-Net · attention mechanism

## 1  Introduction

Medical image segmentation is important for clinical applications, including disease diagnosis, treatment planning, and image-guided interventions. Accurate and efficient segmentation of abdominal organs from medical images is important for assessing organ function, detecting abnormalities, and guiding surgical procedures. However,multi- organ segmentation is a challenging task due to the complex anatomical structures, variabilities in organ shapes and sizes, and the presence of noise and artifacts. Further more, it is difficult to obtain labeled data, unlabeled data is easier to access. In recent years, deep learning based method are widely used for abdominal multi-organ segmentation with good results, among which nnU-Net [7] is one of the most used methods. But nnU-Net's high resource consumption and low inference speed, it does not meet the Challenge's requirements for fast and low-resource. In this work, the main contributions are summarized as follows:

– We use a 2.5D segmentation framework, which can utilize 3D information from CT and does not increasing the computing complexity.
– Introduce an attention mechanism into the U-Net architecture to better capture salient regions and suppress irrelevant background regions.

## 2   Method

### 2.1   Preprocessing

Our method includes the following preprocessing steps:

- **Threshold truncation:**In our opinion, it will encounters difficulties when it comes to segmenting small organs,particularly, more focus is needed on accurately segmenting extremely small organs with unclear boundaries, such as the inferior vena cava (IVC) and duodenum.One possible approach to address this challenge is through threshold value to distinguish the target organ from the surrounding tissues. This technique can help improve the segmentation accuracy for small and indistinct boundary organs.
- **Cropping strategy:**We crop the images and labels based on the slices containing labels and discard the slices without labels. Along the z-axis, we reduce the number of slices to the power of 2 to speed up the subsequent data reading.
- **Resamping method for anisotropic data:**We use this method to resize the slice to reduce GPU memory usage.
- **Intensity normalization method.**

### 2.2   Proposed Method

As shown in figure 1, our method follows the standard U-Net [14] design to achieve the organ segmentation. Specifically, We introduce the attention mechanism into the UNet segmentation network to enhance its ability to focus on region of interest while suppressing irrelevant background region. The attention module can be well embedded in skip connection, which can improve the performance of the model without adding too much computation.In terms of details, we connect three consecutive slices to form a 3D input, which allowing us to fully utilize the 3D information without significantly increasing the GPU memory usage. This approach optimizes memory usage while preserving the spatial context across slices.

  **network architecture details.** Our proposed UNet-CBAM network consists of a combination of UNet network and CBAM module, which consists of spatial attention module and channel attention module. The network input first goes through 5 convolution modules and 4 max pooling layers to complete the downsampling process, and then goes through 5 convolution modules and four upsampling to get the output. The output of each layer of the downsampling path is connected by the features of the skip connection and the upsampling path, respectively. The skip connection performs channel-wise and spatial-wise feature correction on the features throughthe CBAM module.

  **Loss function:** we use the summation between Dice loss and cross-entropy loss because compound loss functions have been proven to be robust in various medical image segmentation tasks [8].

  **Strategies to deal with the partial labels.** The dataset provided by the FLARE 2023 challenge included 2200 CT scans with partial labels, and we did

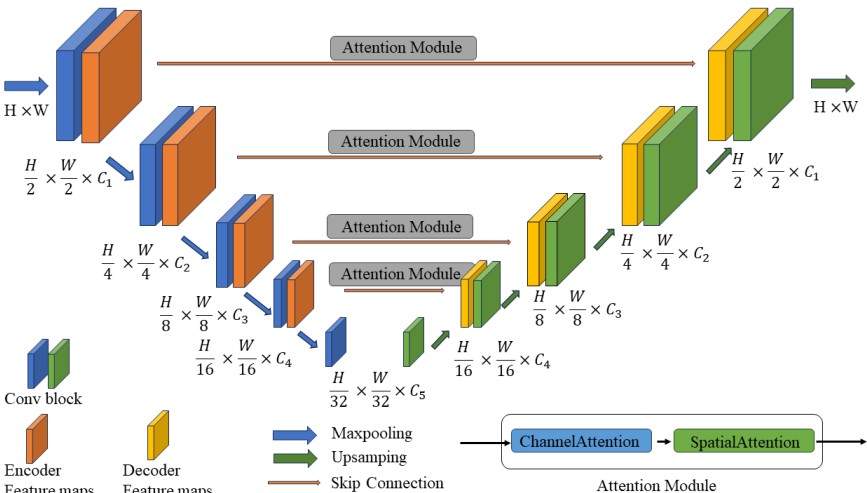

**Fig. 1.** Network architecture

not know which organ was labeled in each case, but due to the amount of the data was sufficient, our method did not make special treatment for the data with partial labels.

**Strategies to use the unlabeled data.** Unlabeled images were not used.

**Strategies to improve inference speed and reduce resource consumption.** We have introduced attention modules in skip connection of U-Net, which can speed up inference and reduce parameters compared to other attention modules. In order to avoid the problem that the loss of 3D information and the low segmentation accuracy of the pure 2D method, we use multiple slices.We also resize the image to reduce resolution to improve inference speed and reduce resource consumption.

### 2.3    Post-processing

We use connected component-based post-process to remove noise and isolated pixels and improve segmentation results.

## 3    Experiments

### 3.1    Dataset and evaluation measures

The FLARE 2023 challenge is an extension of the FLARE 2021-2022 [10][11], aiming to aim to promote the development of foundation models in abdominal disease analysis. The segmentation targets cover 13 organs and various abdominal lesions. The training dataset is curated from more than 30 medical centers under the license permission, including TCIA [2], LiTS [1], MSD [15], KiTS [5,6],

autoPET [4,3], TotalSegmentator [16], and AbdomenCT-1K [12]. The training set includes 4000 abdomen CT scans where 2200 CT scans with partial labels and 1800 CT scans without labels. The validation and testing sets include 100 and 400 CT scans, respectively, which cover various abdominal cancer types, such as liver cancer, kidney cancer, pancreas cancer, colon cancer, gastric cancer, and so on. The organ annotation process used ITK-SNAP [17], nnU-Net [7], and MedSAM [9].

The evaluation metrics encompass two accuracy measures—Dice Similarity Coefficient (DSC) and Normalized Surface Dice (NSD)—alongside two efficiency measures—running time and area under the GPU memory-time curve. These metrics collectively contribute to the ranking computation. Furthermore, the running time and GPU memory consumption are considered within tolerances of 15 seconds and 4 GB, respectively.

### 3.2   Implementation details

**Environment settings** The development environments and requirements are presented in Table 1.

**Table 1.** Development environments and requirements.

| | |
|---|---|
| System | Windows10/Ubuntu 20.04.4 LTS |
| CPU | Intel(R) Core(TM) i9-12900K CPU@3.20GHz |
| RAM | 4×4GB; 2400MT/s |
| GPU (number and type) | One RTX 2080Ti 8G |
| CUDA version | 11.4 |
| Programming language | Python 3.8 |
| Deep learning framework | Pytorch(torch 1.7.0, torchvision 0.8.2) |
| Specific dependencies | medicaltorch,pandas,scipy,collections |
| Code | |

**Training protocols** The Training protocols and details (e.g.,batch size,epoch, optimizer) are presented in Table 2. In the training process,the batch size is 16 and the patch size is fixed as 3*192*192.for optimization, we train it for 150 epochs using Adam with a learning rate of 0.001 and the learning rate reduction strategy using CosineAnnealingLR.

## 4   Results and discussion

### 4.1   Qualitative results on validation set

Figure 2 shows the segmentation results of our method.It clearly illustrates that our method can obtain better segmentation results on large organs than on small organs.However,our segmentation results are clearly missing some organ labels.

**Table 2.** Training protocols.

| | |
|---|---|
| Network initialization | |
| Batch size | 16 |
| Patch size | $3\times192\times192$ |
| Total epochs | 150 |
| Optimizer | Adam |
| Initial learning rate (lr) | 0.001 |
| Lr decay schedule | CosineAnnealingLR |
| Training time | 82 hours |
| Loss function | |
| Number of model parameters | 74.1M[1] |
| Number of flops | 8.22G[2] |
| $CO_2$eq | 1 Kg[3] |

**Table 3.** Quantitative evaluation results. While the proposed method shows promising results in segmenting large organs like the liver, spleen, and kidneys, it still faces significant challenges when it comes to segmenting small organs. Specifically, more attention needs to be paid to extremely small and indistinct boundary organs such as the right adrenal gland (RAG) and esophagus.

| Target | Public Validation | | Online Validation | | Testing | |
|---|---|---|---|---|---|---|
| | DSC(%) | NSD(%) | DSC(%) | NSD(%) | DSC(%) | NSD (%) |
| Liver | 97.66±0.40 | 97.51±1.08 | 83.45 | 83.45 | 94.84 | 93.21 |
| Right Kidney | 97.35±1.92 | 97.19±3.01 | 95.24 | 93.30 | 90.68 | 86.81 |
| Spleen | 98.20±0.11 | 96.69±4.16 | 87.86 | 82.30 | 94.78 | 93.23 |
| Pancreas | 77.38±12.25 | 83.59±13.02 | 72.98 | 81.95 | 75.67 | 83.63 |
| Aorta | 0 | 0 | 14.95 | 15.30 | 13.02 | 12.20 |
| Inferior vena cava | 0 | 0 | 0.00 | 0.00 | 0.00 | 0.00 |
| Right adrenal gland | 0 | 0 | 0.20 | 0.20 | 0.23 | 0.23 |
| Left adrenal gland | 0 | 0 | 1.00 | 1.00 | 0.93 | 0.93 |
| Gallbladder | 45.65±53.28 | 49.46±51.22 | 10.00 | 10.00 | 9.49 | 9.49 |
| Esophagus | 0 | 0 | 0.00 | 0.00 | 0.46 | 0.46 |
| Stomach | 53.56±37.14 | 56.85±37.43 | 14.83 | 16.25 | 9.21 | 8.39 |
| Duodenum | 0 | 0 | 0.00 | 0.00 | 0.00 | 0.00 |
| Left kidney | 95.06±2.13 | 94.42±0.33 | 85.58 | 82.44 | 90.18 | 87.96 |
| Tumor | 0 | 0 | 19.41 | 11.29 | 18.55 | 9.37 |
| Average | 43.45 | 44.29 | 36.83 | 36.68 | 37.03 | 36.81 |

**Table 4.** Quantitative evaluation of segmentation efficiency in terms of the running them and GPU memory consumption. Total GPU denotes the area under GPU Memory-Time curve. Evaluation GPU platform: NVIDIA QUADRO RTX5000 (16G).

| Case ID | Image Size | Running Time (s) | Max GPU (MB) | Total GPU (MB) |
|---------|------------|------------------|--------------|----------------|
| 0001 | (512, 512, 55) | 14.7 | 1570 | 14158 |
| 0051 | (512, 512, 100) | 20.69 | 1570 | 22391 |
| 0017 | (512, 512, 150) | 28.79 | 1570 | 33231 |
| 0019 | (512, 512, 215) | 38.65 | 1570 | 46742 |
| 0099 | (512, 512, 334) | 57.79 | 1570 | 72527 |
| 0063 | (512, 512, 448) | 74.66 | 1570 | 95581 |
| 0048 | (512, 512, 499) | 88.88 | 1570 | 114971 |
| 0029 | (512, 512, 554) | 101.26 | 1570 | 131953 |

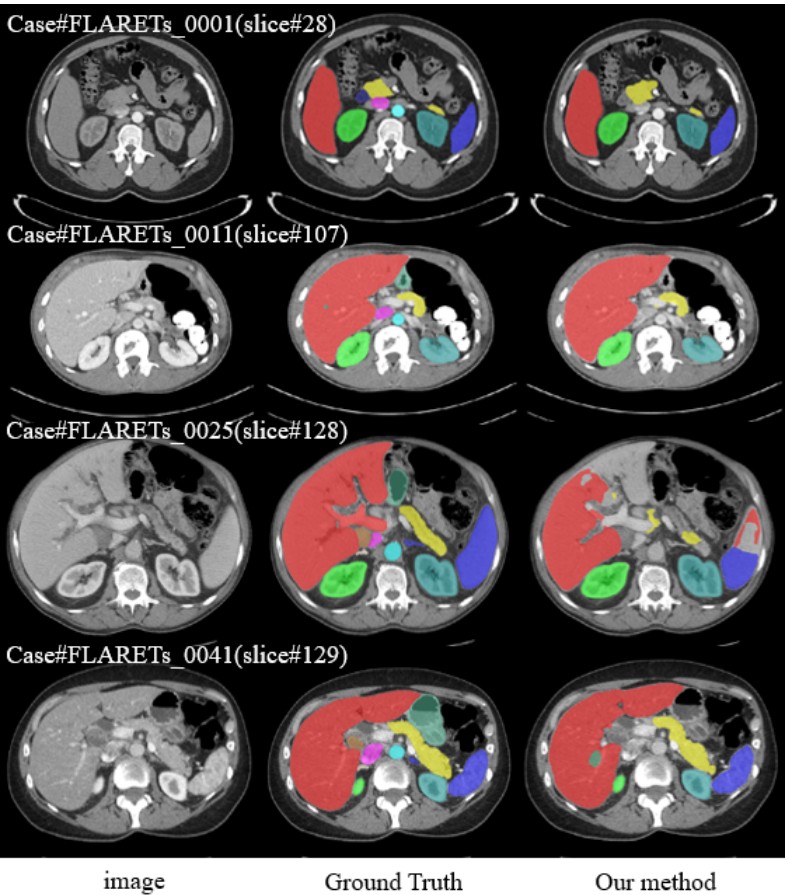

**Fig. 2.** Example cases from the MICCAI FLARE 2023 validation set. Our method does not achieve good segmentation results on the validation set,and here are just two examples that seem to have slightly better segmentation results(No.0001 and No.0011) and two examples that have poor segmentation results(No.0025 and No.0041).

### 4.2    Segmentation efficiency results on validation set

We evaluated the segmentation efficiency on validation set,some of the results are shows in Tabel 4.

### 4.3    Results on final testing set

As shown in Tabel 3, our method achieves a mean DSC of 0.3703 and a mean NSD of 0.3681 on the FLARE 2023 final testing set.

### 4.4    Limitation and future work

Our proposed method for abdominal organ segmentation does not achieve good segmentation results,the limitation of our method is that we are not taking full advantage of unlabeled data,and trained on the data with partial labels may introduce noise and inconsistencies in the training process,leading to reduce model performance.Therefore,we will focus on some techniques such as active learning or semi-supervised learning to iteratively select and annotate the most informative instances,improving the model's performance with partial labeled data.

## 5    Conclusion

In this work,we propose a 2.5D-based U-Net for abdominal multi-organ segmentation,By utilizing partial label data during the training process,we have overcome the challenges of incomplete data annotating.Future research can further extend this method and validate it in a broader range of medical image segmentation tasks.

**Acknowledgements** The authors of this paper declare that the segmentation method they implemented for participation in the FLARE 2023 challenge has not used any pre-trained models nor additional datasets other than those provided by the organizers. The proposed solution is fully automatic without any manual intervention.We thank all the data owners for making the CT scans publicly available and CodaLab [13] for hosting the challenge platform.This study is supported by National Natural Science Foundation of China (62271149),Fujian Provincial Natural Science Foundation project (2021J02019,2021J01578).

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

**Table 5.** Checklist Table. Please fill out this checklist table in the answer column.

| Requirements | Answer |
| --- | --- |
| A meaningful title | Yes |
| The number of authors ($\leq 6$) | 3 |
| Author affiliations, Email, and ORCID | Yes |
| Corresponding author is marked | Yes |
| Validation scores are presented in the abstract | Yes |
| Introduction includes at least three parts: background, related work, and motivation | Yes |
| A pipeline/network figure is provided | Figure 1 |
| Pre-processing | Page 2 |
| Strategies to use the partial label | Page 2 |
| Strategies to use the unlabeled images. | Page 3 |
| Strategies to improve model inference | Page 3 |
| Post-processing | Page 3 |
| Dataset and evaluation metric section is presented | Page 3 |
| Environment setting table is provided | Table 1 |
| Training protocol table is provided | Table 2 |
| Ablation study | Page number |
| Efficiency evaluation results are provided | Table 4 |
| Visualized segmentation example is provided | Figure 2 |
| Limitation and future work are presented | Yes |
| Reference format is consistent. | Yes |