# OpenReview forum: "2.5D U-Net for abdominal multi-organ segmentation"
_MICCAI.org/2023/FLARE — Submitted to FLARE 2023_

### Official Review · Reviewer_k3FV · 2023-09-20
**2.5D U-Net for abdominal multi-organ segmentation**

**Rating:** 5
**Confidence:** 4

**Review:**

Pros:
1. The method introduces an attention mechanism into the U-Net architecture to better
capture salient regions and suppress irrelevant background regions.

Cons：
1. Inconsistent article layout, some punctuation have a space after them while some don't.
2. What do you mean by saying 'we did not know which organ was labeled in each case'? Couldn't you know which organ was labeled by value in labeled file?
3. There is no 'Public Validation' column in Table 3.
4. Please describe about in what kind of cases the proposed method works well (relatively). And Is there any reason for bad preformance besides not using unlabeled data?

---

> ### Author Response · Authors · 2023-11-13
>
> First of all, thank you very much for your valuable comments on the method I proposed.Due to limited time at that time, I did not check the paper format carefully, and the experimental results were not perfect.For poor performance, I think one of the main reasons is because I did not adopt a process from coarse segmentation to fine segmentation.

---

### Official Review · Reviewer_NMt9 · 2023-09-26
**A conventional full supervision solution with just-so-so performance**

**Rating:** 4
**Confidence:** 4

**Review:**

The authors proposed a 2.5D U-Net solution to the MICCAI 2023 FLARE challenge. They employed the U-Net model with attention module and trained it with full supervision. The framework is OK but not novel enough. Moreover, the performance is a little bit poor, especially for small organs.
Besides, the table 3 missed testing results.

---

> ### Author Response · Authors · 2023-11-13
>
> First of all, thank you very much for your valuable comments on the method I proposed.Due to the limited ability and the short time of exposure to abdominal multi-organ segmentation, the experimental results are poor, and I will try to improve it in the future work

---

### Official Review · Reviewer_sWAg · 2023-10-20
**3D segmentation network may help**

**Rating:** 3
**Confidence:** 5

**Review:**

This paper utilizes the conventional 2D U-Net framework with the introduction of an attention mechanism. However, the method is relatively traditional, and its performance is mediocre.

---

> ### Author Response · Authors · 2023-11-13
>
> First of all, thank you very much for your valuable comments on the method I proposed. I will try my best to use a more novel method in my future work

---

### Official Review · Reviewer_kPjA · 2023-10-21
**2.5D U-Net for abdominal multi-organ segmentation**

**Rating:** 4
**Confidence:** 4

**Review:**

Review:

The paper presents a novel approach for abdominal organ segmentation in medical images using the U-Net architecture. The method aims to address the challenges posed by anatomical variations and the proximity of organs in the abdominal region. To enhance segmentation accuracy, an attention mechanism is introduced into the U-Net architecture. This mechanism enables the network to focus on salient regions while suppressing irrelevant background regions, thus improving overall segmentation performance. Additionally, 3D information is incorporated by connecting three consecutive slices as 3-dimensional inputs, allowing spatial context utilization across the slices while minimizing the increase in GPU memory usage.

---

> ### Author Response · Authors · 2023-11-13
>
> Thank you very much for your review and valuable comments.

---

### Decision · Program_Chairs · 2023-10-24

Accept